# Intraoperative Load Sensing in Total Knee Arthroplasty Leads to a Functional but Not Clinical Difference: A Comparative, Gait Analysis Evaluation

**DOI:** 10.3390/jfmk7010023

**Published:** 2022-02-18

**Authors:** Michele Giuntoli, Michelangelo Scaglione, Enrico Bonicoli, Nicola Piolanti, Gianmarco Puccioni, Karlos Zepeda, Emanuele Giannini, Stefano Marchetti, Pier Francesco Indelli

**Affiliations:** 1Department of Orthopaedic Surgery, University of Pisa, Via Paradisa 2, Cisanello, 56124 Pisa, Italy; michelangelo.scaglione@unipi.it (M.S.); enrico.bonicoli@gmail.com (E.B.); nicpio@hotmail.it (N.P.); gm.puccioni@gmail.com (G.P.); stefano.marchetti1@unipi.it (S.M.); 2Harlem Campus, Touro College of Osteopathic Medicine, Harlem, New York, NY 10027, USA; kzepeda@student.touro.edu; 3Kinetic Center, 56124 Pisa, Italy; kineticenterlab@gmail.com; 4Department of Orthopaedic Surgery, Stanford University School of Medicine, Stanford, CA 94305, USA; pindelli@stanford.edu

**Keywords:** TKA, total knee arthroplasty, knee, kinematics, biomechanics, load sensor, gait analysis, knee balance, sensor, mid-flexion instability

## Abstract

Introduction: Although Total Knee Arthroplasty (TKA) is a successful procedure, a significant number of patients are still unsatisfied, reporting instability at the mid-flexion range (Mid-Flexion Instability-MFI). To avoid this complication, many innovations, including load sensors (LS), have been introduced. The intraoperative use of LS may facilitate the balance of the knee during the entire range of motion to avoid MFI postoperatively. The objective of this study was to perform a Gait Analysis (GA) evaluation of a series of patients who underwent primary TKA using a single LS technology. Methods: The authors matched and compared two groups of patients treated with the same posterior stabilized TKA design. In Group A, 10 knees were intraoperatively balanced with LS technology, while 10 knees (Group B) underwent standard TKA. The correct TKA alignment was preoperatively determined aiming for a mechanical alignment. Clinical evaluation was performed according to the WOMAC, Knee Society Score (KSS) and Forgotten Joint Score, while functional evaluation was performed using a state-of-the-art GA platform. Results: We reported excellent clinical results in both groups without any statistical difference in patient reported outcome measurements (PROMs); from a functional standpoint, several GA space–time parameters were closer to normal in the sensor group when compared to the standard group, but a statistically significant difference was not reached. Conclusions: Gait Analysis represents a valid method to evaluate TKA kinematics. This study, with its limitations, showed that pressure sensitive technology represents a valid aid for surgeons aiming to improve the postoperative stability of TKA; however, other factors (i.e., level of intra-articular constraint and alignment) may play a major role in reproducing the normal knee biomechanics.

## 1. Introduction

Total Knee Arthroplasty (TKA) is a successful surgical procedure for the treatment of symptomatic, end-stage knee osteoarthritis (OA) and its use has rapidly increased over time. However, a significant number of patients still report not been fully satisfied with their TKA, even when the components seem to be well-positioned radiographically. Changes in the knee proprioception and subtle instability have been advocated as main reasons for this “unsatisfaction” rate; in particular, instability at the mid-flexion range, generally called “Mid-Flexion Instability” (MFI), has been often quoted by researchers as an undesired symptom following TKA [1,2,3].

In order to solve this problem, surgeons, as well as the industry, propose continuous innovations: on the surgeon side, the classical dogma pursuing a mechanical alignment during TKA has been revised in favor of a more kinematic approach; on the industry side, new component designs have been introduced to the market, few of them having a femoral, multi-radius sagittal design, an asymmetric tibial component and modular polyethylene inserts characterized by a different level of constraint within the same system. Moreover, new technical tools have been developed, generically called Computer Assisted Surgery (CAS), including navigation, robotics, custom-made cutting guides, and load sensors [4,5].

Load sensors represent an interesting technology, since they have an affordable cost, are quickly available, require a minimum set-up in the operating room, and allow to objectively quantify the intra-articular balance at different degrees of knee flexion. However, their actual effectiveness is still debated and most of the studies currently available have short follow-ups (FUs) [6,7,8]. In addition, some authors have questioned the cost/benefit ratio of these devices [9,10,11].

Gait Analysis showed to be an accurate and non-invasive tool to study in vivo joints kinetic and kinematic during the range of motion (ROM) [12,13,14]. In the TKA world, it has been used to evaluate the in vivo kinematic of the knee during the Load-Response (LR), Mid-Stance (MS) and Swing phases of gait.

The aim of this study was to determine the utility of a load sensing intraoperative technology in reproducing a more natural gait after TKA, considering its cost, and limited functional outcome data available in the literature. The author’s hypothesis was that the use of this technology would improve the postoperative knee kinematics when evaluated with Gait Analysis. In this paper, we report a single-center experience on the intraoperative use of pressure-sensitive sensors to balance TKAs: a series of patients who received a sensor-balanced TKA were clinically and functionally compared, using Gait Analysis, to a group of patients who received a standard performed TKA.

## 2. Materials and Methods

Between January 2019 and January 2020, 137 patients (137 knees) underwent TKA at the first author’s institution. In this consecutive series, the Legion posterior-stabilized (PS) TKA design (Smith & Nephew, London, UK) was implanted in 76 patients: in 32 patients (Group A), joint balancing was performed with the use of the load pressure system VERASENSE^TM^ (Orthosensor Inc., Dania Beach, FL, USA), while in the remaining 44 patients (Group B), the balancing was performed according to a conventional technique. Twenty patients from Group A and twenty patients from Group B were matched for sex, age, and BMI (Body Mass Index): ten patients from each group agreed on participating in the current study, which was performed during the COVID-19 pandemic. The final study population consisted of 20 patients, 10 for each group: in Group A (sensor group), 4 males and 6 females were included; in Group B (traditional group), 4 males and 6 females participated.

The main inclusion criteria were patients with symptomatic knee osteoarthritis grade III or IV according to Kellgren and Lawrence [15], aged between 65 and 80 years at the time of surgery, had the ability to walk without crutches after TKA, had a postoperative neutral lower limb alignment (with a tolerance of 3 degrees with respect to the mechanical axis) and consented to participate to the current study.

The main exclusion criteria were patients who were unable to walk without crutches, those who had Parkinson’s disease or other neuro-muscular conditions influencing a normal gait, previous surgery in the lower limbs and BMI over 35. All surgical procedures were performed by the same surgeons following an identical hybrid (measured resection in extension/gap balancing in flexion) surgical technique [16]: the same, cemented, posterior-stabilized (PS) TKA design was utilized in both cohorts (Legion PS, Smith & Nephew, London, UK) [17].

The Oxinum^TM^ Legion postero-stabilized (PS) femoral component is characterized by having a sagittal J curve design and asymmetrical conformation of the posterior condyles on the frontal plane; its tibial component is titanium made and it has an asymmetrical conformation, allowing for better matching of the patient’s tibial anatomy and for an easier to achieve rotational alignment.

Accurate preoperative planning was always performed by measuring, on the standing calibrated X-rays, the amount of bony resection from the distal femur and proximal tibia according to the desired alignment. Consequently, a calipered measurement of the bone cuts has always been intraoperatively performed. The preoperative alignment was determined on lower limb weight-bearing X-rays by measuring the mechanical femoral–tibial angle (mFTA), the angle between the femoral and tibial mechanical axes measured at the medial side of the lower limb, while the patella’s high and sagittal slope was evaluated on lateral view X-rays. In case of a severe varus or valgus knee, we preferred not to completely correct the deformity according to the mechanical alignment, but we chose a “restricted kinematic” alignment to improve the knee proprioception [16,18].

The VERASENSE^TM^ system consists of a wireless polyethylene insert trial equipped with baroceptors able to detect and record the load at the femoral–tibial interface during trialing and after placements of the final components. This device can discriminate the medial compartment pressure from the lateral one, highlighting a possible major difference in the inter-compartmental pressure. Intraoperatively, the device is positioned into the knee during passive ROM testing and the detected data are displayed on a tablet in the operative room. In addition to reporting on the intra-articular load, the pressure sensor can also provide data about the spatial localization of the femoral–tibial contact point and its displacement during flexion–extension. Measurements are typically taken at 10°, 45° and 90° of knee flexion [19]. In order for the knee to be defined as “balanced”, according to the current literature [20] and as suggested by the manufacturer’s user guide, it must be balanced both in the sagittal (antero–posterior) and coronal (medio–lateral) planes. The current authors considered “well-balanced” to be those knees which demonstrated an intraoperative stable endpoint during the posterior drawer test and did not show an elevation of the tibial trial (a sign of excessive tightness of the flexion gap) or a “paradoxical motion” (anterior translation of the femur, sign of looseness of the flexion gap) during flexion. A satisfactory coronal balance, on the other hand, was defined as a pressure delta between the two compartments of ≤15 pounds (approximately 6.8 kg) throughout the whole passive ROM intraoperative testing; this value was arbitrarily chosen according to the literature [20,21]; at this point, accessory soft tissue releases and/or the bone recuts were performed according to the load reading (Figure 1).

All patients underwent, preoperatively and at the latest FU, clinical evaluation according to the Western Ontario and McMaster Universities’ Arthritis Index (WOMAC) [22] and the Knee Society Score (KSS), which is subdivided into Knee Score (KS) and Function Score (FS) sections [23]. The Forgotten Joint Score-12 (FJS) [24] was also administered to the patients at the time of the final FU.

All patients were functionally evaluated at the final FU with Gait Analysis. This evaluation was performed using the BTS GAITLAB system, provided by BTS Bioengineering (Garbagnate Milanese, Milan, Italy). This gait technology consists of an optoelectronic system with six high-frequency infrared cameras (BTS SMART DX with a sensor resolution of 2.2 Mpixel, acquisition frequency at maximum resolution of 340 fps, maximum acquisition frequency of 2000 fps and an accuracy/volume < 0.1 mm on 4 × 4 × 3 mt.) capable of detecting the reflective markers placed on the patient’s landmarks and of two force platforms capable of measuring the GRF (Ground Force Reaction) vectors acting on the patient’s joints. The reflective markers were positioned on the patient landmarks according to the Helen Heyes Medial Marker protocol [25,26,27] (Figure 2).

The obtained data were reprocessed with a dedicated software (BTS GAIT LAB) after entering the patient’s anthropometric parameters. A detailed report, containing average space–time parameters, kinematic standing angles, qualitative walking indexes (Gait Profile Score—GPS; Gait Variable Scores—GVSs; Gait Deviation Index—GDI), the outcome of kinematic and dynamic analysis, was automatically calculated by the same software. For each measured parameter, statistical analysis was carried out using T-tests and U-tests (Mann–Whitney test).

### 2.1. Space–Time Parameters

The gait space–time parameters evaluated in this study were the following: cycle duration (s); support duration (s); swing duration (s); support phase (%); oscillation phase (%); single support phase (%); double support phase (%); average speed (m/s); average speed (% height/s); cadence (steps/min); cycle length (m); cycle length (% height); step length (m) and step width (m). Both knees (affected and normal, contralateral) were evaluated but we analyzed and reported only the data related to the operated limb.

### 2.2. Gait Profile Score (GPS), Gait Variable Scores (GVSs) and Gait Deviation Index (GDI)

The average GPS and GDI were compared between the operated limbs in patients of Group A (sensor group) and Group B (traditional group). The average GVSs were obtained for the operated limb of each patient from both groups to evaluate which parameters had the greatest influence on the GPS.

### 2.3. Kinematic Analysis

A bilateral knee kinematic descriptive analysis was carried out in each patient; particular attention was paid to the average flexion–extension and internal–external rotation of the knee and internal-external rotation of the ipsilateral hip during the contact/LR phase (0–10% of the gait cycle) and during the intermediate stance phase (10–30% of the gait cycle). The average internal–external rotation of the hip in the Swing phase (60–100% of the gait cycle) and the average flexion peak during the whole gait cycle were also statistically evaluated.

### 2.4. Kinetic Analysis

A descriptive analysis of the dynamic parameters of both limbs was carried out for each patient comparing the obtained diagrams, while a statistical analysis between the operated limbs of the patients belonging to the two groups was applied to the 1st and 2nd peak of the adduction moment of the knee (Peak KAM—Knee Articular Moment—1st and 2nd) and at the average flexion–extension moment of the knee (KFM—Knee Flexion Moment) in the Mid-Stance phase (MS, 0–10% of the gait cycle). The peak of the Knee Flexion Moment (Peak KFM) in the Heel Strike/Load-Response phase (HS/LR, 0–10% of the gait cycle) was also statistically evaluated.

## 3. Results

All 20 patients originally enrolled in the study were available at a minimum FU of 9 months: 4 males and 6 females were included in Group A (sensor group), while 4 males and 6 females were included in Group B (traditional TKA group).

### 3.1. Radiographic and Clinical–Functional Results

The mean age at the time of surgery was 70.6 years (SD: 4.68/range: 67–78 years) in Group A (sensor group) and 71.5 years (SD: 4.74/range: 66–78 years) in Group B. The mean post-surgical FU was 9.86 months (SD: 2.5/range: 6–14 months) in Group A and 9.88 months (SD: 3.15/range: 6–14 months) in Group B (Table 1).

In the preoperative radiographic evaluation, the mean medial femoral–tibial mechanical angle (mFTA) was 171.5° (SD: 4.15/range: 166–179°) in Group A and 173.2° (SD: 3.84/range: 172–178°) in Group B; at the last FU, this angle showed an average value of 178.9° (SD: 0.99/range: 177–180°) in Group A and 179° (SD: 1.06/range: 178–181°) in Group B (Table 2).

The mean duration of surgery was 121 min (SD: 10.84/range: 105–130) in Group A and 97 min (SD: 12/range: 80–110) in Group B, with an average increase of 24 min in Group A; this was statistically significant.

In Group A (sensor group), the mean FJS-12 score was 74.40 (SD: 21.15/range: 35.4–96.83) at the latest FU; the mean preoperative WOMAC score was 52.26 (SD: 11.47/range: 42–61), while the mean WOMAC score at the latest FU was 10.53 (SD: 10.71/range: 4–32). The mean preoperative clinical KSS was 41.37 (SD: 10.2/range: 15–60) and the mean preoperative functional KSS was 38.6 (SD: 12.2/range: 15–55); the mean latest KSS scores were 94.24 (SD: 12.1/range: 65–100) and 92.5 (SD: 13.88/range: 60–100) for the clinical KKS and functional KSS, respectively. In Group B (no sensor group), the mean FJS-12 score at the latest FU was 67.15 (SD: 26.12/range: 10.5–91.7); the mean preoperative WOMAC score was 45 (SD: 9.64/range: 39–56), while the mean score at the latest FU was 14.77 (SD: 10/range: 4–31). The mean preoperative clinical KKS was 49.5 (SD: 13.45/range: 25–60) and the mean preoperative functional KSS was 44.65 (SD: 9.46/range: 20–65); the mean KSS scores at the latest FU were 92 (SD: 11.44/range: 70–100) and 87.77 (SD: 17.15/range: 50–100) for clinical KKS and functional KSS, respectively (Table 3 and Figure 3).

Both groups significantly improved with respect to their preoperative status, with Group A achieving a better score; however, the statistical comparison between the analyzed outcomes did not show a significant difference (*p* < 0.05) between the two groups.

### 3.2. Gait Analysis Results

Data analysis regarding the space–time parameters showed no statistically significant differences between the two groups (Table 4) at final FU. Data analyses related to the Gait Profile Score (GPS) and Gait Deviation Index (GDI) are reported in Table 5; we did not identify any statistically significant differences.

The kinematic parameters relative to the operated limbs in the two groups of patients are reported in Table 6, the comparison between the two groups showed a weak significance (*p* < 0.05 on the T-test, but not strong in the U-test) in knee flexion during the intermediate stance phase (MS), which was greater in Group B patients (no sensor group), and a certain trend (*p* < 0.15 on the T-test) in extra-rotation of the hip, which was greater in Group A patients (sensor group) during the intermediate support phase.

The descriptive analysis related to GVSs showed how the parameters that most influenced the GPS results for Group A (sensor group) were excessive internal rotation of the hip, excessive flexion of the hip and excessive dorsiflexion of the ankle; on the other hand, in Group B, excessive dorsiflexion of the ankle, excessive internal rotation of the hip and excessive knee flexion all characterized the gait patterns. Nonetheless, these differences were only trends, since no statistically significant differences were demonstrated during GVS analysis.

The kinetic parameters of this study are reported in Table 7. The kinetic comparison between the two groups showed a weak significance (*p* < 0.05 at T-test, but not significant at U-test) in the mean extensor moment of the knee at the MS phase (Mid-Stance KEM), lower in Group A and in the peak of the Knee Flexion Moment at the LR phase (Peak KFM LR), which was higher in Group A. There was also a difference, not statistically significant, in the second peak of the adductor moment of the knee (Peak KAM2 Abd/Add), which was slightly greater in the patients of Group A.

## 4. Discussion

The ultimate aim of this study was to evaluate, through Gait Analysis, if the intraoperative use of load sensors was correlated with a clinical and functional improvement in the final outcome. Interestingly, the hypothesis of this study was not confirmed since the authors were not able to find a statistically significant increase in clinical outcomes when the load sensors were intraoperatively used. On the other hand, the authors demonstrated an improvement in all clinical scores in both study groups between preoperative and the latest evaluation. Moreover, a greater, not statistically significant, functional increase was found in the sensor group compared to the conventional technique group.

Since its introduction as a surgical procedure, TKA has been continuously improved leading to a progressive increase in functional outcomes and patient satisfaction. These promising results have determined the extension of this surgical solution to younger osteoarthritic patients having greater functional demands. In the last few decades, however, despite numerous technological advances (new implant designs, new surgical tools and modern approaches), patient satisfaction has not increased as expected [28]. The recent literature still reports that 20% of patients are dissatisfied with their knee replacements, despite the radiological evaluation of their implants routinely confirming the excellent positioning of the implants [7,28]. In the last few years, mid-flexion instability, defined as joint instability at intermediate degrees of flexion (between 15° and 60°), has been addressed as one of the main causes of implant failure; being more a symptom that a clinical finding, instability is increasing as a cause for TKA revision. Therefore, the goal of obtaining an articular kinematic similar to the native knee has been stated as essential to obtain better clinical and functional results [1,2,3]. Achieving this goal is extremely demanding, since the TKA procedure involves the sacrifice of important anatomical structures, which cannot be easily reproduced into the implant; modern prosthetic designs and surgical techniques must consider these needs and ensure good joint mobility, maintaining, at the same time, an optimal load distribution at the prosthesis/cement/bone interfaces. The latest industry innovations (surgical navigation, robotics, augmented reality and pressure sensors) point in this direction.

Historically, an adequate joint balance was considered crucial to reproduce optimal knee kinematics: correct balancing represents one of the most difficult goals to achieve during a TKA procedure, as it is often entrusted to the surgeon’s preferences and experience. Patients’ factors and comorbidities, such as high BMI, the presence of severe deformities, and the presence of fixed contractures or ligament laxity, all could make it challenging to obtain an adequate balance of the knee, especially at intermediate degrees of flexion [7] because of the role of the dynamic knee stabilizers.

A direct correlation between inaccurate, intraoperative mid-flexion joint balance and postoperative mid-flexion instability has been hypothesized by many authors [1,2,29]. To facilitate this surgical step, the use of load sensor devices, able to provide quantitative data in real time about forces and load distribution between the tibial and femoral components throughout the whole range of motion, has been introduced as a valid tool to correct soft tissue balancing and consequently to achieve better clinical and functional results. Moreover, load sensors, avoiding an overload on the polyethylene insert, promise a lower wear rate and therefore a greater survival of the implant.

The current literature is scant regarding studies analyzing the influence of load sensors on helping patients to regain appropriate knee proprioception and kinematics when functionally tested using Gait Analysis. Few authors have reported significantly higher clinical and functional outcomes and higher satisfaction after load-sensor balanced TKA when compared to the conventional technique, while other reports have shown similar results, highlighting multiple limitations of this load sensor technology [8,10,30,31,32,33]. Nodzo et al. have criticized the use of the VERASENSE system because they found a statistically significant change in the lateral compartment readings between the pressure measurement taken with the trial components and the final implant [34], making reproducibility a major issue. Similarly, Chow et al. suggested a significant change in pressure values between the measurement made with the trial components and the one obtained after cementing the final components [35]. Finally, Nicolet-Petersen et al. suggested that different sequential readings are obtained when the device is not reset after an initial, compartmental overload (>70 pounds) [36].

Differently from the study conducted by Chow et al. [35], in our series, a load pressure check with the definitive components was not systematically performed. Conversely, we occasionally performed a load check with the definitive components on-site, and we did not highlight excessive variations with respect to the measurements carried out with the trial components.

In respect to previous studies, our case series presents a limited number of patients, due to the recent introduction of VERASENSE in the leading institution country market (2017), the cost of the device, and the possibility to use the device with a single manufacturer’s implant (Legion PS, Smith & Nephew, London, UK); these represent major limitations of our study. Furthermore, the small cohort study was largely influenced by the ongoing COVID-19 pandemic, which limited the possibility to evaluate all the patients originally enrolled in the study [37]. The reduced number of patients, in particular, limited the possibility of achieving a statistically significant result when the two cohorts were compared. Interestingly, the average increase in operating time, when the sensor technology was used, made, in our opinion, the benefit/risk ratio advantageous.

In this study, the functional evaluation using Gait Analysis provided a large amount of data referable not only to the affected knee but also to the homolateral joints and the contralateral limb. The most significant finding of the current study was that no statistically significant differences in the space–time parameters in the gait cycle were found between the two groups of patients, making the systematic use of this sensor technology not recommended. The data regarding GPS and GDI, an estimate of global walking function, indicated no statistically significant differences between the two groups. The GVSs, an estimate of the parameters that most influence GPS, indicated that the deviation from normal values was mainly due to rotation of the femur, hip flexion, and ankle movements in both groups. Therefore, the space–time parameters and the global scores of the Gait Analysis indicated no major differences between the groups. The comparison with the contralateral limb, however, did not show major differences for each individual patient, demonstrating the good overall results obtained with the TKA procedure, independently from the use of the sensor.

The kinematic analysis showed that knees balanced with load sensors presented less knee flexion during the intermediate stance phase than patients balanced with the conventional technique, while the external femoral rotation during the intermediate support phase was greater in the load sensor group (Figure 4), the current authors previously hypothesized that an inferior knee flexion during the intermediate support phase indicated a better implant balance and ultimately greater joint stability [12].

The results of the current study are in line with those reported in the literature when PS TKA knees were evaluated by Gait Analysis; the finding of a greater external femoral rotation during the intermediate support phase in the sensor group was originally reported by Andriacchi et al. [13] in a similar Gait Analysis study. Regarding kinetic parameters, the current authors found a higher average KEM (Knee Extension Moment) at the Mid-Stance phase and a higher peak of KFM (Knee Flexion Moment) at the load-Response phase of the gait in the sensor group when compared to the conventional group. Our interpretation of these sagittal plane kinetic data was that the finding of a reduction in the knee flexion during the MS phase in the sensor group could indicate a better mid-flexion balance with optimization of the quadriceps femoral lever arm [12]. The current authors hypothesized that this trend in the improvement of muscle recruitment was also facilitated by the use of a PS design, which ultimately posteriorizes the femoral–tibial contact point; the overall, not statistically significant but still present as a trend, greater muscular efficiency in the sensor group could ultimately improve the stability of the joint, as shown by Ghirardelli et al. [12].

Dennis et al. showed a reduction in the articular motion and sagittal joint moment following TKA, independently from the level of intra-articular constraint [14]. Limitation of joint mobility up to a real stiffness represents a well-known strategy adopted by patients with advanced knee OA [38]. In fact, it has been previously shown [38,39,40,41] that, by limiting knee flexion, the magnitude of the flexion–extension moment decreases and consequently, the intensity of activation of the quadriceps femoris, which must support a reduced force vector. Since the leverage of the quadriceps femoris is determined by the patellar action at the time of decreased muscle contraction, the extensor mechanism and the patella do not “overstuff” the femoral–tibial joint, having, as a direct consequence, a reduction in pain on the joint; this mechanism has been previously defined as “quadriceps avoidance” [38,39,40,41]. Interestingly, it has been shown [38,39] that patients often maintain this “quadriceps avoidance” attitude even after TKA surgery, despite an improvement in the level of pain. On the other hand, the kinematic goal of a PS TKA is to favor an increase in the quadriceps lever arm, improving its efficiency. In this scenario, the quadriceps muscle could better absorb the KFM externally applied in the LR phase to ultimately limit the joint excursion in the subsequent stance phase, guaranteeing the patient greater safety and stability in the single support phase of the gait.

Another finding of the current study was that the comparison of the second peak of KAM between the two groups of patients did not demonstrate a statistically significant difference: in fact, the flexion–extension and abduction–adduction moments during the gait cycle were similar in both groups (Figure 5).

The current authors hypothesized that the reduction of these external forces highlighted in both groups of TKA patients was due to the identical surgical technique and the implant design, ultimately ensuring less stress and therefore greater longevity to the implant. The current authors are aware that, to better understand the actual ability of patients to react to the ground force and to respond to the external moments applied, an electromyographic analysis during the gait data acquisitions could be preferable; this represents another limitation of our study. In fact, an electromyographic analysis could have determined the muscle’s activation during the subsequent stages of the gait, ultimately validating or confuting our hypothesis. Conversely, it has been shown that space–time parameters can influence the joint angular movement and moments in the lower limb [42,43]; greater stride length and walking speed could increase joint ROM, while higher speed could eventually increase the forces applied to the ground and consequently increase joint moment.

Finally, our study confirmed that Gait Analysis represents an optimal methodology to study TKA patients, although more needs to be studied and understood [44]; a better understanding could be obtained when the gait cycle is assessed globally, given the large number of variables involved.

The poor statistical value of our work, a final major limitation of the current study, could be attributed to the small cohort of patients studied, although this type of study is usually applied to small groups of patients. Furthermore, in our study, patient selection (i.e., presence of major hip and ankle pathologies) may have influenced the results. On the other hand, the health emergency related to COVID-19, which decisively influenced the final phase of this work, prevented us to have a larger sample.

## 5. Conclusions

The current study, despite its multiple limitations related to the small number of patients and the short follow-up, reported excellent clinical results in both groups of patients and better functional results in the sensor patient group, despite not being statistically significant. Gait Analysis has proven to be a valid, but complex method to study in vivo TKA kinematics. The results obtained, although with a relative statistical value, have globally demonstrated a better reproduction of knee kinematics in the sensor patient group, especially in the HR, LR, and MS gait phases, guaranteeing stability at the mid-flexion range. Future studies should investigate the application of electromyography during Gait Analysis testing in order to quantify the quality and quantity of muscle activation in TKA patients. This report, to our knowledge, represents the first study evaluating the influence of the use of a load sensor technology on patients’ gait. From this perspective, the data obtained here could represent a good starting point for future studies, with much larger cohorts and more selective inclusion criteria.

## Figures and Tables

**Figure 1 jfmk-07-00023-f001:**
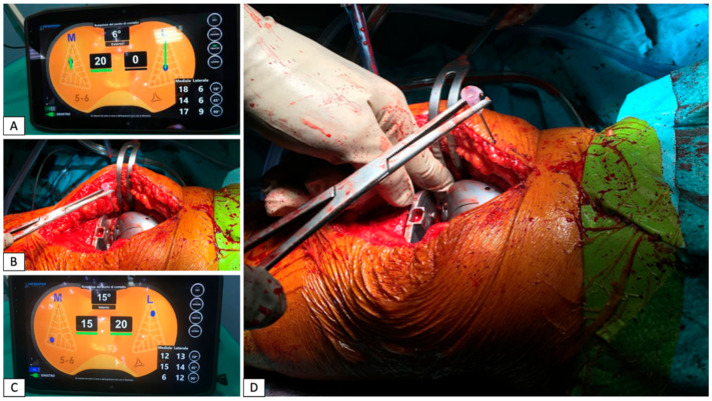
The first measurement detected with the trial components did not result in an optimal balance: the medial compartment was too tight while the lateral one was too loose (**A**). The surgeon tried to achieve the right balance between medial and later compartment by performing a release of the medial compartment using the “pie crust” technique (**B**). The ligament tension detected after the medial release has been considered satisfying, with an equal forces distribution among the two compartments of the knee (**C**). In a severe varus or valgus knee, according to measured resection, a medial or lateral ligamentous release can be performed to obtain a better balance: the “pie crust” technique consists of performing small cuts in the too tight structures using an 18-gauge needle or a n. 11 scalpel blade (**D**).

**Figure 2 jfmk-07-00023-f002:**
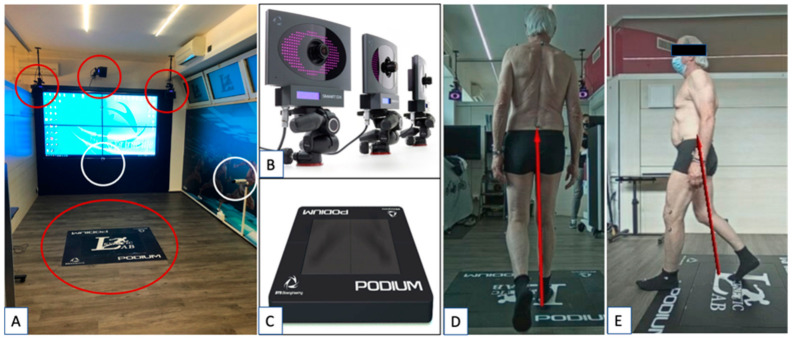
(**A**) The gait lab set-up at the Kinetic Center (Pisa, Pi, Italy) with the infra-red cameras and the two Podium force platforms—red circles; (**B**) details of optoelectronic infra-red cameras capable of detecting the reflecting markers on to the patients; (**C**) details of one Podium platform used to measure the GFR during a patient’s walk. The cameras, the platforms and the data software employed were provided by BTS Bioengineering (Garbagnate Milanese, Mi, Italy). In (**D**,**E**), a patient performs a test in the gait lab: one can notice the markers placed on the patient according to Helen Heyes Medial Marker protocol and the GFR (red vectors). The GFR vectors can be identified by the cameras in white circles in A when the patient engages one of the two Podium force platforms.

**Figure 3 jfmk-07-00023-f003:**
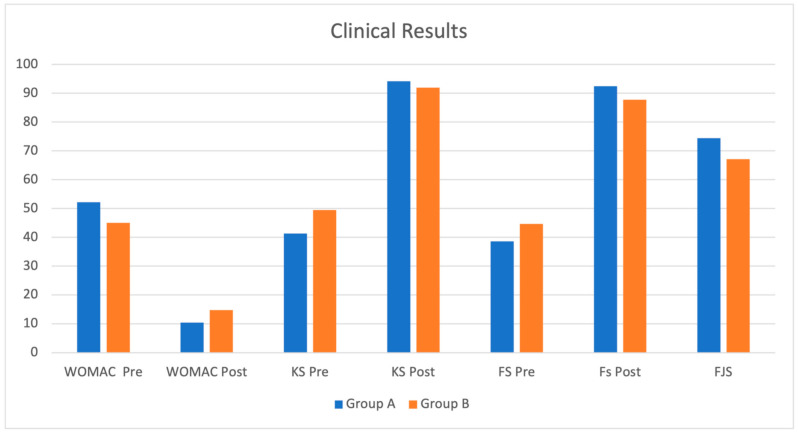
Clinical results for the two study groups. No significant differences in clinical improvement among the two studied groups of patients were found, although both groups showed an improvement in clinical function. However, Group A (sensor group) patients seemed to have slightly better clinical outcomes than Group B.

**Figure 4 jfmk-07-00023-f004:**
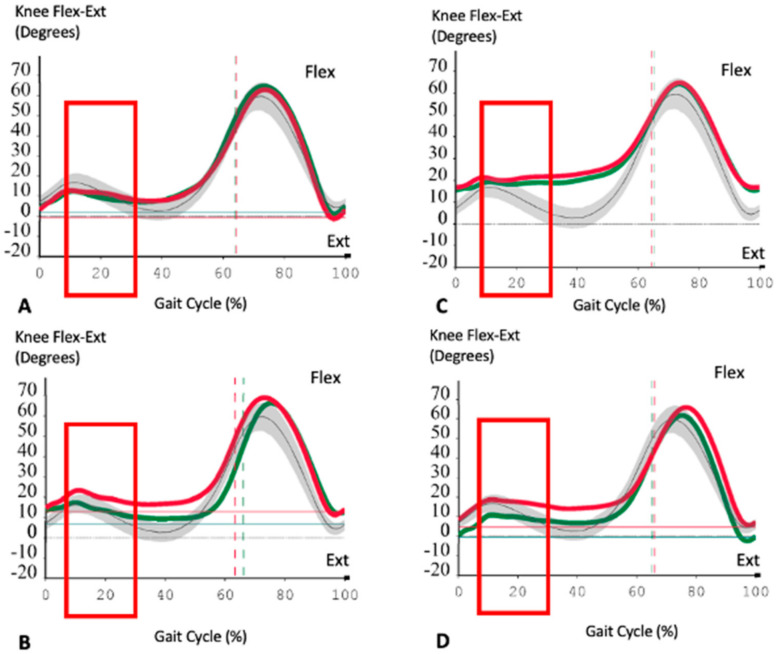
Descriptive kinematic analysis of 4 patients: two from Group A (**A**,**B**) and two from Group B (**C**,**D**). In (**A**,**B**), the operated knee is the right one (green lines) while in (**C**,**D**), the operated knee is the left one (red lines). Normal values are represented by grey lines and the grey area (SD). Red rectangles refer to the specific Gait Phase (MS). Minus values on the ordinate line refer to hyper extension degrees of the knee. We can notice how the operated knee flexion, at the MS phase (10–30% gait cycle), is more similar normal values in the prosthetic knee balanced with load sensor devices than the TKA balanced with standard technique. Another interesting finding is the similar behavior between the two knees of the same patient (prosthetic and non-prosthetic one). This “twin” pattern is globally detectable in all gait measurements.

**Figure 5 jfmk-07-00023-f005:**
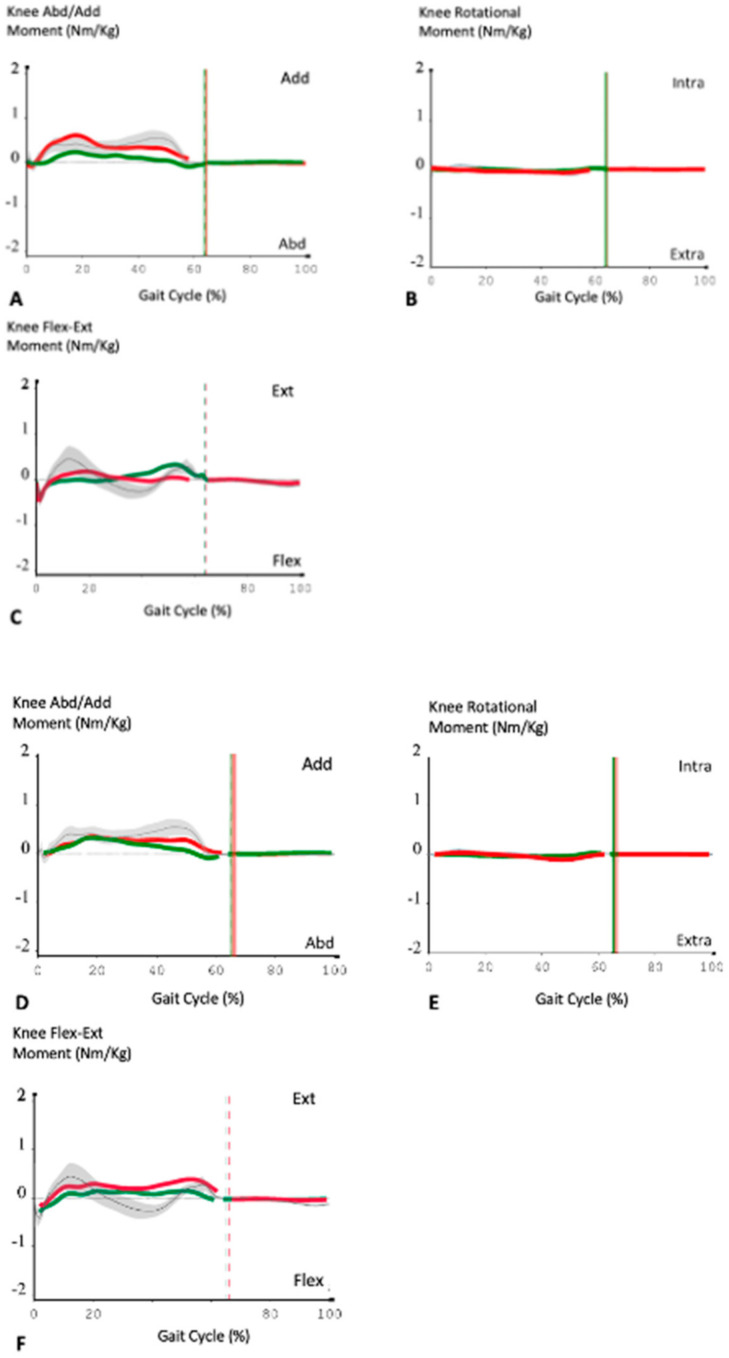
Descriptive kinematic analysis of two patients: one from Group A (**A**–**C**) and one from Group B (**D**–**F**). In both series, the operated knee is the right one (green lines) and normal values are represented by grey lines and the grey area (SD). The red lines represent the left, non-operated, knee. The flexion–extension, abduction–adduction and the rotational moments of the prosthetic knee had no major variations during the gait cycle. A reduction of these external forces could be related to the surgical technique used (mechanical alignment) and the type of prosthesis employed (PS one). It would ensure less stress and therefore greater longevity to the implant.

**Table 1 jfmk-07-00023-t001:** Demographic parameters of study population.

Demographic Parameters	Group A (Mean ± SD)	Group B (Mean ± SD)
Age (y.o.)	70.6 ± 4.68	71.5 ± 4.74
Post-Surgical FU (months)	9.86 ± 2.5	9.88 ± 3.15

**Table 2 jfmk-07-00023-t002:** Preoperative and postoperative lower limb mechanical alignment for each group of studied patients.

Medial Mechanical FemoralTibial Angle (mFTA)	Preoperative (Mean ± SD)	Postoperative (Mean ± SD)
Group A (degree)	171.5 ± 4.15	178.9 ± 0.99
Group B (degree)	173.2 ± 3.84	179 ± 1.06

**Table 3 jfmk-07-00023-t003:** Clinical results for the two study groups.

Clinical Score	WOMAC		KSS (KS/FS)		FJS-12
	Preoperative	Last FU	Preoperative	Last FU	Last FU
Group A (mean ± SD)	52.26 ± 11.47	10.53 ± 10.71	41.37 ± 10.2/ 38.6 ±12.2	94.24 ± 12.1/ 92.5 ±13.88	74.40 ± 21.15
Group B (mean ± SD)	45 ± 9.64	14.77 ± 10	49.5 ± 13.45/ 44.65 ± 9.46	92 ± 11.44/ 87.77 ±17.15	67.15 ± 26.12

**Table 4 jfmk-07-00023-t004:** Space–time parameters. No significant variation was found between Group A and Group B patients.

Space–Time Variables	Group A (Mean ± SD)	Group B (Mean ± SD)
Cycle Duration (s)	1.27 ± 0.08	1.25 ± 0.06
Stance Duration (s)	0.84 ± 0.05	0.83 ± 0.04
Swing Duration (s)	0.43 ± 0.04	0.43 ± 0.03
Stance Phase (%)	66 ± 1.18	66.04 ± 1.67
Swing Phase (%)	34.01 ± 1.45	34.27 ± 1.32
Single Support Phase (%)	34 ± 2.00	34.19 ± 1.03
Double Support Phase (%)	15.43 ± 1.75	16.18 ± 1.35
Average Speed (m/s)	0.78 ± 0.15	0.75 ± 0.13
Average Speed (%height/s)	44.62 ± 6.94	46.29 ± 8.73
Cadence (step/min)	94.65 ± 5.37	95.51 ± 4.43
Cycle Length (m)	0.99 ± 0.23	0.90 ± 0.10
Cycle Length (%height)	57.14 ± 1.13	57.6 5± 8.62
Step Length	0.49 ± 0.14	0.45 ± 0.05
Step Width	0.09 ± 0.04	0.09 ± 0.08

**Table 5 jfmk-07-00023-t005:** GPS and GDI qualitative walking indexes. No significant variation between Group A and Group B patients was found; even globally for both TKA patients, we did not find any value different from normal values.

	Group A (Mean ± SD)	Group B (Mean ± SD)	N.V.	Test—T	Test—U (Mann–Whitney)
Gait Profile Score (GPS)	9.57 ± 1.72	9.27 ± 0.85	<7	NS	NS
Gait Deviation Index (GDI)	82.04 ± 7.41	81.59 ± 3.48	>100	NS	NS

**Table 6 jfmk-07-00023-t006:** Most significant kinematic parameters for both study groups. We report a weak significance in knee flexion during the intermediate stance phase (MS), which is greater in Group B patients (no VERASENSE), and a certain trend (*p* < 0.15 on the T-test) in extra-rotation of the hip, which is greater in patients of the Group A (VERASENSE) during the intermediate support phase. LR: Load-Response; MS: Mid-Stance.

Kinematic Angles (Degrees)	Group A (Mean ± SD)	Group B (Mean ± SD)	Test—T	Test—U (Mann-Whitney)
Knee Flex-Ext LR	10.42 ± 4.23	15.85 ± 7.71	NS	NS
Knee Intra-Extra LR	3.07 ± 10.56	−5.8 ± 12.86	NS	NS
Hip Intra-Extra LR	−7.72 ± 8.82	−12.75 ± 5.04	NS	NS
Knee Flex-Ext MS	9.15 ± 4.16	17.65 ± 6.23	*p* < 0.05	NS
Knee Intra-Extra MS	3.5 ± 11.31	−5.1 ± 11.52	NS	NS
Hip Intra-Extra MS	−17.47 ± 5.05	−10.97 ± 5.19	*p* = 0.07	NS
Knee Flexion Peak	67.72 ± 1.47	64.02 ± 4.85	NS	NS

**Table 7 jfmk-07-00023-t007:** Most significant kinetic parameters for both study groups. We report a weak significance in the mean extensor moment of the knee at the MS phase (Mid-Stance KEM), lower in Group A, and in the peak of the flexion moment of the knee at the LR phase (Peak KFM LR), which was higher in Group A. There was also a difference, not statistically significant, in the 2nd peak of the adductor moment of the knee (Peak KAM2 Abd/Add), which was slightly greater in the patients of Group A. LR: load-Response; MS: Mid-Stance; KAM: Knee Adduction Moment; KEM: Knee Extension Moment.

Kinetic Parameters (Nm/Kg)	Group A (Mean ± SD)	Group B (Mean ± SD)	Test—T	Test—U (Mann-Whitney)
Peak KAM2 Abd/Add	0.45 ± 0.18	0.3 ± 0.07	*p* = 0.13	NS
Mid-Stance KEM	0.01 ± 0.09	0.23 ± 0.06	*p* < 0.05	NS
Peak KFM LR	−0.46 ± 0.09	−0.16 ± 0.05	*p* < 0.05	NS

## Data Availability

Data are available at first author’s department.

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
