# Peer review of "Intraoperative Load Sensing in Total Knee Arthroplasty Leads to a Functional but Not Clinical Difference: A Comparative, Gait Analysis Evaluation"

_jfmk, 2022, doi:10.3390/jfmk7010023_

Round 1

Reviewer 1 Report

This study contained a newly device to detect medial -lateral balancing in the TKA, which tell us ML balance in any knee range of motion. The effectiveness of these devices for walking ability was investigated to compare with them to without them using gait analysis. But, significant difference was not detected in the study except some special cases. This study is interesting for knee surgeons including me. However, several points to correct in the manuscript.

  1. Table 1,2,3, Figure 3 are same in the text, respectively. Duplication should be avoided.
  2. Table 2; preoperative FTAs were less than 180 degrees. In your patients, most of them were valgus knee? In general, knee deformity is varus in many patients.
  3. Corrections; P1,L21 Gait Analysis (GA); P5,L163 was pro-; P9,L265 Mead------Mid; P15,L498 Journal-----J; P15,L526 Bone Joint J; P15,L528 Journal-----J; P15,L542 Ann Rheum Dis ; P15,L544 J Orthop Surg Res ; P16,L568 J Clin Orthop trauma

Author Response

1) Q: Table 1,2,3, Figure 3 are same in the text, respectively. Duplication should be avoided.

A: We agree with the reviewer: any duplication should be avoided. However, we decided to add a graphical representation of PROMs without numbers to make its interpretation for the readers easier; on the other hand, many readers appreciate details and a tabular representation with data might be easier to interpretate than numbers only in the results section.

2) Q: Table 2; preoperative FTAs were less than 180 degrees. In your patients, most of them were valgus knee? In general, knee deformity is varus in many patients.

A: This clarification is correct. In our series, like in most of the arthritic population, we registered a varus pre-operative deformity as dominant. When reporting FTA, we referred to the mechanical axis measured to the medial aspect of the lower limbs, so’, for example, an angle less than 180 degrees implies a varus deformity, while an angle over 180 degrees implies a valgus deformity. We specified this measurement technique in the body of the revised manuscript (lines 107-109, lines 220-224, Table 2) to reduce the possibility of confusing the reader.

3) Q: Corrections; P1,L21 Gait Analysis (GA); P5,L163 was pro-; P9,L265 Mead------Mid; P15,L498 Journal-----J; P15,L526 Bone Joint J; P15,L528 Journal-----J; P15,L542 Ann Rheum Dis ; P15,L544 J Orthop Surg Res ; P16,L568 J Clin Orthop trauma.

A: Thank you for your comment. We have revised the text and added your suggested correction to the revised manuscript.

Reviewer 2 Report

Improving the results of TKA is based on a better understanding of the issues that are not (yet) addressed by surgery and implant designs. This is very well-acknowledged by the authors. Load Sensors (LS) are only one of many small additional tools that may improve our understanding and our surgical technique. Gait analysis offers valuable information in addition to PROMs and clinical and radiology. The topic is of major interest.

The patient's number in the two groups might be insufficient and is a weakness of this study, but this is correctly acknowledged by the authors.  

The results did not show a better result for the load sensor group, probably because surgery was performed by an experienced surgeon. A similar study,  with surgery performed by a less experienced surgeon, might be interesting.  

Regarding References: 

line 549-550-551  please check - the authors and the article are not in continuity 

Author Response

A: line 549-550-551 please check - the authors and the article are not in continuity.

Q: We agree with the Reviewer. Thank you for your comment. We have included the correction in the revised manuscript.